# Retinal Toxicity Induced by Chemical Agents

**DOI:** 10.3390/ijms23158182

**Published:** 2022-07-25

**Authors:** Daniel Souza Monteiro de Araújo, Rafael Brito, Danniel Pereira-Figueiredo, Alexandre dos Santos-Rodrigues, Francesco De Logu, Romina Nassini, Andrea Zin, Karin C. Calaza

**Affiliations:** 1Department of Health Sciences, Clinical Pharmacology and Oncology Section, University of Florence, 50139 Florence, Italy; daniel.souzamonteirodearaujo@unifi.it (D.S.M.d.A.); francesco.delogu@unifi.it (F.D.L.); 2Department of Cellular and Molecular Biology, Institute of Biology, Universidade Federal Fluminense, Niterói 24210-201, RJ, Brazil; rafaelbrito@id.uff.br; 3Department of Neurobiology, Institute of Biology, Universidade Federal Fluminense, Niterói 24210-201, RJ, Brazil; dannielfigueiredo@id.uff.br (D.P.-F.); alexandre_rodrigues@id.uff.br (A.d.S.-R.); kcalaza@id.uff.br (K.C.C.); 4Clinical Research Unit, National Institute of Women, Children and Adolescents Health, Fernandes Figueira, FIOCRUZ, Rio de Janeiro 22250-020, RJ, Brazil; andrea.zin@iff.fiocruz.br; 5Brazilian Institute of Ophthalmology, Rio de Janeiro 22250-040, RJ, Brazil

**Keywords:** pesticides, medicinal herbs, natural products, medicinal, retina, cell death, neurotoxicity

## Abstract

Vision is an important sense for humans, and visual impairment/blindness has a huge impact in daily life. The retina is a nervous tissue that is essential for visual processing since it possesses light sensors (photoreceptors) and performs a pre-processing of visual information. Thus, retinal cell dysfunction or degeneration affects visual ability and several general aspects of the day-to-day of a person’s lives. The retina has a blood–retinal barrier, which protects the tissue from a wide range of molecules or microorganisms. However, several agents, coming from systemic pathways, reach the retina and influence its function and survival. Pesticides are still used worldwide for agriculture, contaminating food with substances that could reach the retina. Natural products have also been used for therapeutic purposes and are another group of substances that can get to the retina. Finally, a wide number of medicines administered for different diseases can also affect the retina. The present review aimed to gather recent information about the hazard of these products to the retina, which could be used to encourage the search for more healthy, suitable, or less risky agents.

## 1. Introduction

At least 2.2 billion people have vision impairment and in approximately 1 billion of these cases, vision impairment could have been prevented or has not yet been addressed. Although most people affected are over the age of 50 years, vision loss can occur with people of all ages, with the incidence increasing with age. People with visual deficits must overcome a series of barriers imposed by our way of life. For children, visual impairment represents a risk of social exclusion and isolation [1].

The retina is responsible for light transduction and for preprocessing this information with an intraretinal circuitry. The mature vertebrate retina is located at the back of the eye, and it is organized in layers [2,3]. In the outermost portion of the retina, closer to retinal pigmented epithelium (RPE), lies the outer nuclear layer (ONL) formed by the cell bodies of cones and rods photoreceptors; the next nuclear layer, toward vitreal surface, is the inner nuclear layer (INL), composed of the cell bodies of horizontal, bipolar, amacrine, displaced ganglion cells, and in some species, interplexiform cells. Closest to the vitreal part of the retina, there is the ganglion cell layer (GCL) consisting of the cell bodies of ganglion cells, displaced amacrine cells, and a few photosensitive ganglion cells. Interspersing these nuclear layers are the outer plexiform (OPL) and inner plexiform (IPL) layers, where synaptic contacts occur. The axons of ganglion cells give rise to the optical fiber layer (OFL), becoming the optic nerve as these fibers leave the retina, sending the information to areas of the brain responsible for visual processing [2,4].

In the retina, specialized cells develop relevant functions in visual processing, with cone cells working in daylight conditions (photopic), whereas rods function in dim light conditions (scotopic) [5]. Photoreceptors synapse with specific rod- or cone-bipolar cells and horizontal cells in the OPL. Each bipolar cell, in turn, synapses with ganglion cells. Therefore, the transduced light information travels through the excitatory vertical/radial pathway [2,6], which is modulated by horizontal and amacrine cells, respectively in the OPL and IPL, by inhibitory synapses composing the horizontal/lateral pathway.

Besides neurons, the retina also presents three types of glial cells: Müller, astrocyte, and microglia. The Müller glia is the predominant glial cell type (90% of glial cells), with the cell body located in the middle of the INL and radial processes that extend through all layers of the retina. Müller glia perform several functions crucial to the homeostasis and physiology of the retina, including pro-survival signaling, synaptic regulation, neuronal activity modulation, extracellular potassium buffering, water elimination, regulation of blood–retinal barrier (BRB) and neurovascular coupling [7,8,9,10]. Astrocytes are restricted to the innermost part of this tissue, in the OFL and GCL. At these sites, astrocytes have a neurotrophic support role, interact with blood vessels participating in the BRB assembly and neurovascular coupling, and can react to pathological signs by adopting a gliosis pattern [11,12]. Microglia are mononuclear phagocytes that enter the retina during development [13,14]. Although microglia can be sparsely found in almost all retinal layers, the great majority resides in the OPL and IPL [14]. Microglia are also important players in the maintenance of the retinal homeostasis, responding to neural signs that switch the microglia state (in a simplistic way from ramified vigilant to ameboid reactive cell), probably contributing to retinal visual function [15,16,17].

Most mammalians have intraretinal blood vessels supporting the energy demands of the inner retina. Human retina has a peripapillary plexus in the innermost part of the NFL and inner (in the GCL) and outer (in the IPL, INL to OPL) intraretinal beds. Interestingly, the primate specialized region for high visual acuity, the fovea, is avascular, minimizing light distortion [18]. Similar to the brain, intraretinal vessels have a specialization, tight junctions between the endothelial cells, that guarantees retinal protection, and it is called BRB. Finally, the source of blood supply for the outer retina, the photoreceptors layer, is the choroidal vascularization, which is vast but has no BRB.

Between the retina and choroid stands the RPE, which plays several important roles to support retinal function. RPE is a monolayer of pigmented epithelial cells which absorb unprocessed light, contributing to a high visual acuity. RPE also contributes to the recycling of molecules of the visual cycle and to the transport of oxygen and nutrients coming from choroidal vessels to the photoreceptor. Finally, RPE forms an outer BRB through tight junctions present between RPE cells, and malfunctioning/degeneration of RPE is associated to diseases such as age-related macular degeneration (AMD) and Stargardt disease [17,19].

A wide variety of substances, such as quinolines, phenothiazines, and antiretroviral drugs reach the retina through the vascular supply. While beneficial therapeutically, several ocular and systemic drugs are responsible for causing retinal damage. Usually, toxicity is reversible following discontinuation of the provoking drug. Nevertheless, permanent or progressive vision loss may occur in a few cases.

Besides medicines, other substances can also reach the retina and can influence retinal function and health. Several chemical agents used as pesticides are hazardous to health, and specifically to the retina.

Between 2005 and 2012, 90% of pesticides used in the United States were restricted to the agriculture sectors, where herbicides accounted for 50% of the total use in 2011 and 60% in 2012 [20]. Brazil is one of the largest consumers of pesticides in the world, along with the United States and China. In 2020, according to the Brazilian Institute for the Environment and Renewable Natural Resources (IBAMA), Brazil sold almost 685 tons of pesticides, a number that is reflected in the estimated consumption of 7.2 L of pesticides per Brazilian, according to the dossier of the Brazilian Association of Collective Health (ABRASCO). Between 2007 and 2014, there were 25,000 cases of intoxication with agricultural pesticides in Brazil and 1186 deaths resulting from this contamination [21], but these numbers may be even higher, due to unreported cases [22]. In addition, the current federal government has adopted a permissive policy on the use of pesticides in Brazil, and 80% of pesticides authorized in the country are banned in at least three countries of the Organization for Economic Cooperation and Development (OECD) of the European Community [22]. Thus, it is very likely that there will be an increase in the number of cases and deaths associated with occupational or occasional exposure in Brazil.

Individuals who are exposed either chronically or acutely can experience side-effects that include cancer, endocrine disruption, reproductive effects, liver and kidney damage, birth defects, and neurotoxicity [22]. In the latter case, both the central nervous system (CNS) and the peripheral nervous system are affected, and a dataset has now extended this effect to the retina, a tissue belonging to the CNS. Here, we present data on the main pesticides used, including chlorpyrifos, cypermethrin, lefenuron, thiamethoxam, glyphosate, triphenyltin, thiram, benomyl, and carbendazim.

Finally, natural products have been used as a source of treatment of several diseases both by empirical/traditional knowledge or referenced by science and used in health systems. Most of the published data show protective roles in distinct pathological conditions. In the present review, data from substances belonging to this class of chemical agents that are harmful to the retina were included: Kava kava extract, hypericin, *Embelia ribes*, and *Hagenia abyssinica*.

Therefore, this review aimed to gather information related to substances, mainly pesticides, natural products, and drugs, which threaten retinal function and health (refer to Appendix A to see details about the search method).

## 2. Pesticides

### 2.1. Organophosphates

#### 2.1.1. Chlorpyrifos

Chlorpyrifos (O,O-diethyl-O-3,5–6-trichloro-2-pyridyl-phosphorothioate, CPF) is an insecticidal pesticide that belongs to the organophosphate (OP) class. It is commonly used by farmers to combat crop pests, although it is present in many household insecticides. In 2012, CPF was the fourteenth most widely used pesticide in the United States, with an average usage of between 4 million to 8 million pounds [20]. In Brazil, in 2020, CPF was the tenth most-sold active ingredient, totaling 8864.88 tons, according to IBAMA [21]. Also in 2020, the European Commission decided to ban the use of this pesticide due to concerns related to human health, such as a possible genotoxicity and developmental neurotoxicity. As OP, the primary action of chlorpyrifos is to inhibit the enzymatic activity of acetylcholinesterase (AChE), the enzyme responsible for metabolizing acetylcholine into choline and acetic acid [23]. Because of this effect, there is an overstimulation of cholinergic neurotransmission. Exposure and absorption of this pesticide occurs through the respiratory system, gastrointestinal tract and skin, and the most affected systems are the cardiovascular, respiratory, and nervous systems. Actions in the CNS include neuropathological, neurophysical, neurobehavioral, and neurochemical changes [23].

Epidemiological evidence has shown that farmer pesticide applicators for OP-based insecticides had a higher incidence of age-related macular degeneration [24], while licensed pesticide applicators had a higher prevalence of retinal degeneration [25]. These findings in humans show that chlorpyrifos acts as a harmful agent to the retina and studies carried out in animal models have corroborated the effect. The primary action of CPF in inhibiting the enzymatic activity of AChE was demonstrated in retinas of rats submitted to chronic treatment (6 and 12 months) with daily doses of this pesticide (1 mg/kg/day and 5 mg/kg/day) associated or not with an episodic dose every two months (65 mg/kg and 45 g/kg) [26,27].

The alteration of AChE functionality has implications for retinal physiology, as dark-adapted mice that received the episodic dose of CPF during chronic treatment had a slower recovery of sensitivity after exposure to bright light, measured by electroretinogram [27]. Although other neurochemical parameters have not been investigated by the authors, the inhibition of AChE contributes to the effect, even though no morphological changes were found in the retina of these animals nor changes in the density of acetylcholine muscarinic receptors [26,27] (Table 1 and Figure 1).

CPF can promote cell damage through oxidative stress. Kunming mice administered acutely by gavage with CPF at a concentration of 63 mg/kg (¼ of the lethal dose) showed intense cell death by apoptosis in retinal neurons. The treatment induced an increase in malondialdehyde (MDA) levels, a marker of lipid peroxidation, and a reduction in the activity of antioxidant enzymes, such as superoxide dismutase (SOD), catalase (CAT), and glutathione peroxidase, which indicates an imbalance between the production of reactive oxygen species and cellular antioxidant defenses after CPF administration. Oxidative stress and cell death were inhibited in animals pre-treated for 6 days with a combination of vitamin C (250 mg/kg) and vitamin E (150 mg/kg), two components with antioxidant characteristics [28]. Therefore, CPF-induced cell death was promoted by oxidative stress. However, it is impossible to determine whether this accumulation of ROS arises from CPF metabolism itself and/or whether it comes from functional changes in mitochondria [23]. On the other hand, CPF also inhibited AChE activity and increased intracellular calcium levels, both effects being blocked by vitamin C and E [28], further evidence that ROS production is the primary cause of these effects. The data together reveal that the death mechanisms triggered by CPF reflect different actions acting together or not. For example, AChE is expressed only in retinal amacrine, displaced amacrine and/or bipolar cells depending on the species [29,30], which restricts cell death induced by the inhibition of this enzyme to mostly neighboring cells. In turn, the authors considered a total increase in calcium in the retina, without specifying the cell type. CPF could be increasing calcium, in this context, in only some cell populations, and these cells could also be undergoing cell death through calcium-dependent mechanisms [23,31]. In any case, it is evident that the effects are ROS-dependent and add more information to the neurotoxic action of CPF on the retina.

CPF is also responsible for the increased production of ROS in human retinal pigment epithelial cells 19 (ARPE 19 cells). ARPE 19 cells acutely (24 h) or chronically (9 days) exposed to increasing concentrations of CFP (1 nM to 100 µM) showed an increase in ROS production and a reduction in glutathione levels, with no change in cell viability. In addition, after treatment with CPF, an increase in mRNA and changes in the activity of the enzyme paraoxonase 2 (PON2) was found, a protein related to antioxidant defenses. The resistance to cell death of these cells appears to depend on PON2, as CPF triggered cell death (20%) in cultures whose PON2 expression was ablated [32]. Now, more data need to be gathered to see if PON2 would have any neuroprotective effect on retinal neurons exposed to CPF.

Continuous exposure to CPF also reduces anterograde axonal transport of optic nerve projections towards the superior colliculus of rats. Rats were administered subcutaneously with a single dose of CPF (18 mg/kg) or daily with a dose of 3 mg/kg or 18 mg/kg for 14 days [33]. Treatment with a single dose did not alter AChE activity or modify anterograde transport to the superior colliculus. On the other hand, repeated treatment with CPF reduced enzyme activity by 40% (3 mg/kg) and 80% (18 mg/kg), as well as weakening anterograde axonal transport to the superior colliculus. Fifteen days after the end of treatment, a reduced activity of AChE was still found at the concentration of 18 mg/kg and a weakening in axonal transport at both concentrations [33]. The data make it evident that the CPF acts on the molecular machinery responsible for the anterograde vesicular transport from the retina to the superior colliculus. In this sense, the authors hypothesize direct actions of CPF on the kinesin motor protein and/or on the dynamic instability of microtubules. It has already been shown that CPF and other organophosphate pesticides favor the disruption of the link between kinesin and microtubule, which would, in theory, lead to the interruption of kinesin-dependent vesicular transport in microtubules [34]. At the same time, CPF decreased tubulin polymerization in vitro assays [35,36] and reduced microtubule-associated protein 2 (MAP-2)-protein immunoreactivity in vitro organotypic slice cultures of hippocampus mice [36]. MAP-2 proteins are microtubule accessory proteins responsible for their polymerization and stability. Furthermore, CPF and CPF oxon, an active CPF metabolite, have been shown to covalently bind tubulin and tubulin-associated proteins [37]. Thus, the reduction in CPF-induced axonal transport can be explained by these changes in the microtubule and correlate with the neurobehavioral deficits observed in individuals exposed to OP-based pesticides.

The CPF applied in crops is an environmental contaminant of great proportions, acting in different ecosystems, such as the marine. The CPF can reach the sea through rivers, estuaries, floods, sewage, and rain. Although the effects of CPF on the CNS of marine fauna are scarce, there are studies that have already revealed deleterious effects on the retina of fish. Histopathological examination of the retina of *Mugil cephalus*, *Chanos chanos*, and *Later calcarifer* fingerlings submitted to chronic treatment for 30 days with sublethal doses of CPF showed, depending on the concentration, detachment of the pigmented epithelium from the photoreceptor layer, detachment of the photoreceptor layer from the inner nuclear layer, shrinkage and detachment of the outer plexiform layer from the outer nuclear layer, fusion of photoreceptor cells, and formation of vacuoles in the ganglion cell layer [38,39]. The results suggest extensive cell death in the retina of these animals, as shown in rodent models treated with CPF [28], whereas they contrast with the work with rodents that showed that chronic treatment with CPF does not alter the cytoarchitecture of the retina [26,27]. These differences may be related to the concentrations used, the time of administration, and the physiological characteristics of each animal. However, more research needs to be carried out in the area to corroborate this hypothesis.

#### 2.1.2. Glyphosate

Glyphosate is a widely used herbicide that controls broadleaf weeds and grasses. It has been registered as a pesticide in the United States since 1974. Glyphosate is the most-used herbicide in the United States’ agricultural sector since 2001 [20] and the most-used herbicide in Brazil. In 2020 in Brazil, glyphosate was the most-sold active ingredient, reaching the mark of 246 tons sold according to IBAMA [21], which classifies the pesticide as a Class III agent (Product Dangerous to the Environment). It is an organophosphorus compound that acts by inhibiting the enzyme 5-enolpyruvylshikimate-3-phosphate synthase, which is produced only by plants and microorganisms. This enzyme is essential in the biosynthesis of the aromatic amino acids phenylalanine, tyrosine, and tryptophan in algae, higher plants, bacteria and fungi [40]. The inhibition of this enzyme by glyphosate shuts down the pathway, resulting in organism death due to lack of aromatic amino acids crucial to survive [40]. Using the zebrafish toxicity model system, one group reported that glyphosate (50 µg/mL for 19h) induced several morphological changes in brain architecture, including loss of delineated brain ventricles and reductions in cephalic and eye regions. They detected a different array of eye alterations, such as microphthalmia, and by in situ hybridization analysis, decreases in genes involved in the eye development including pax2, pax6, otx2, and ephA4. This same report also showed a significant decrease of zn-8 (a marker of optic nerve, optic chiasm, and the bifurcation of the optic nerve arisen from the retinal ganglion cells) in glyphosate-treated embryos [41].

### 2.2. Pyrethroid

#### Cypermethrin

Cypermethrin([cyano-(3-phenoxy-phenyl)methyl]3-(2,2-dichloroethenyl)-2,2-dimethylcyclopropane-1-carboxylate, CYP) is a synthetic class II compound of pyrethroids widely used in the field and in the domestic environment in the fight against pests. It is already known that CYP induces macrophage death by arresting the cell cycle and producing ROS [42] and possesses teratogenic effects in rats due to its ability to cross the blood–placental barrier [43]. Little is known about the effects of CYP on the retina. The first finding in mammals refers to rats that were exposed during the gestational period (7th day of gestation to birth) to CYP (12 mg/kg/day). Animals that were exposed during pregnancy to CYP showed marked retinal histopathological differences 7 days and 14 days after birth when compared to control animals. There was a great vacuolization of the inner and outer plexiform layers, appearance of pyknotic nuclei in the nuclear layers and vasodilation and congestion of blood vessels. CYP treatment also increased immunostaining for collagen IV and caspase 3, suggesting a proliferative effect on blood vessels and cell death, respectively. Interestingly, administration with N-acetylcysteine (NAC, 1 g/kg/day) together with CYP prevented the deleterious actions of the pesticide on the retina [44]. The tissue modifications found in the retina are likely related to oxidative stress triggered by CYP, as already shown for macrophages. Thus, the protection achieved by the administration of NAC can be explained by its ability to act as a precursor of glutathione, a molecule that acts as a reducing agent and is one of the main molecules of cell antioxidant defenses. However, it has already been shown that NAC protects macrophages from CYP-induced cell death through the inhibition of c-Jun amino-terminal kinases (JNK) and extracellular signal-regulated (ERK) protein kinases [42]. Whether this pathway is involved in the death of retinal neurons is still unclear. As the production of ROS favors the proliferation of pathological vessels in the retina, the reduction of collagen IV labeling promoted by NAC possibly occurs due to its antioxidant property. 

The findings in mammals were corroborated in the zebrafish model. Zebrafish exposed for 12 days to CYP (0.6 µg/L) showed cells with apoptotic aspects in the ONL and INL, as detected by toluidine blue staining. Indeed, CYP increased labeling for caspase 3 (ONL) and histone γ-H2AX (ONL and INL), a marker of DNA damage. DNA damage may be due to increased ROS, although the treatment with CYP also induced an increase in SOD and CAT activity, maybe as a consequence of intracellular pathways activated by oxidative stress [45]. Together, the data provide evidence that the cell death mechanism behind CYP depends on ROS production, but the details of this process remain to be clarified.

### 2.3. Benzoylurea

#### Lefenuron

The agricultural sector uses several chemical classes of insecticides. However, little is known about the neurotoxic actions of these compounds on the retina, even though acute and chronic contamination in occupational activities is a reality for field workers. In addition, contamination of food and aquatic environments can reach the final consumer, who also uses classes of insecticides to combat domestic pests. Specific studies with insecticides have sought to highlight the effects of this acute and chronic exposure on the morphology and physiology of the retina. Soares and colleagues working with Tambaqui (*Colossoma macropomum*), a fish from Amazonian rivers, demonstrated toxic effects of lefenuron, an insecticide widely used in Brazil. Lefenuron is a benzoylurea that inhibits chitin synthesis in insects, although it is used to eliminate ectoparasites in fish. The authors demonstrated that acute treatment (96 h) with lefenuron (0.7 mg/L and 0.9 mg/L) led to hemorrhages (hyphema) in the animals’ eyes. Exposure to lefenuron, under these conditions, still contributed to changes in the electroretinogram of fish. An increase in the implicit time of a-wave and b-wave was observed. In fish exposed to a concentration of 0.1 mg/L of lefenuron, the b-wave amplitude was increased by 3-fold compared to the control. Interestingly, no changes were found in the electroretinogram in animals submitted to chronic treatment (4 months) with lefenuron (0.1 and 0.3 mg/L) [46]. The authors correlate this change in the electroretinogram in acute treatment with the vascular hemodynamic changes found in the gills. On the other hand, they speculate that the absence of electroretinogram changes in chronic treatment is due to retinal neurogenesis and/or biochemical compensations.

### 2.4. Neonicotinoid

#### Thiamethoxam

Another pesticide used in agribusiness with described effects on the retina and eye structure is thiamethoxam (THIA), a second-generation neonicotinoid insecticide. Its deleterious action on the visual system has been described in *Drosophila melanogaster* in larval stages and in adult individuals. The ocular discs of third instar larvae treated with THIA for 24 h in food (6.75 ug/mL) appeared to be damaged and with a large mass of cells in apoptosis. In turn, in adult flies, THIA (1.04 µg/mL) induced a disorganization of the compound eyes of these animals after 24 h of treatment. Together these data show that acute exposure to sublethal doses of THIA is capable of inducing cell death and disorganizing the ocular structure in Drosophila [47].

### 2.5. Organotin

#### Triphenyltin

Fungicides are another class of pesticides commonly used in agriculture, but unfortunately some of them can be harmful to humans and other animals. Even though some of them are forbidden to be used by some countries and by the International Maritime Organization, because of their capability for persistence and bioaccumulation, such as in the case of triphenyltin (TPT), some countries as Brazil still use this molecule in some farms [48]. Even though the prohibition by some countries, Brazil is not the only one that continues to use this molecule; other countries like the US and China also use it in their agriculture [20,49]. It is known that TPT can act as an agonist of retinoid X receptors [50], and there is evidence that it can cause impairments on the morphogenesis of optic cup, leading, for example, to eyelessness and small eyes of embryos after 5 weeks of 1.6 to 1000 ng/L TPT exposure [51] and impairment in the development of lens and retina after exposure to concentrations going from 0.9 to 18.2 µg/L [52] on fish. Using an *in ovo* nano-injection, Xiao showed that the drug can also disrupt the development of zebrafish retinal axons even at low doses such as 0.8 ng/g of TPT [53]. Although the exposure does not represent real environment conditions, this data adds information regarding TPT toxicity. Taking this into account, another work exposed the eggs directly to TPT in the environment, using concentration levels according to the literature. They observed that the gene expression involved in retinal development and GH/IGF axis was affected after exposure to TPT (1, 10, and 100 ng/L), besides the decrease in survival and hatching rate, body weight and length [54].

### 2.6. Organosulfur

#### Thiram

Another commonly used fungicide is thiram, that is used to prevent fungal diseases in seed and crops and similarly as an animal repellent to protect fruit trees and ornamentals from damage by rabbits, rodents and deer. Among its toxic effects, it has been observed to have an adverse effect on the reproductive activity of Sprague–Dawley CD rats after 13 weeks of daily dose (0, 30, 58, or 132 mg/kg) administration [55], as well as cytotoxicity on Chinese hamster ovary (CHO) cells treated for 18 h showing a big toxicity with a LC50 of 5 × 10^−7^ M [56]. In a long period of exposure to this drug (0.4, 4, and 40 mg/kg/day for 104 weeks), besides the noxious effects such as death of animals during the trials, nausea, vomiting, and clonic seizures, the animals had ophthalmologic damage such as retinal detachment, hemorrhage in fundus of the eye, and miosis [57].

### 2.7. Benzimidazoles 

#### Benomyl and Carbendazim

Another group of fungicides, members of the benzimidazole family, includes benomyl and carbendazim, which have been used in many countries for years. They apparently have a low acute oral toxicity, but if administered in high doses can have some toxic effects. In 1990 a work showed that animals fed (375, 755, and 1500 mg/kg of benomyl and 247, 484, and 969 mg/kg of carbendazim) with those drugs had an accumulation of them in the retina [58].

## 3. Natural Products

The research on these substances has been increasing through the years as many groups report the possibility of using isolated molecules or extracts from plants to treat or to prevent certain diseases. We searched for data suggesting that the usage of these compounds could directly affect the retina or as a side effect of using it to treat another condition. It must be stated that the massive number of results indicated that many compounds have protective effects both in vitro, using cultures of photoreceptors, retinal pigmented epithelial cells, or retinal vessel endothelial cells, or in the retina of a variety of animal models for retinal diseases, such as AMD, retinal ischemia, diabetic retinopathy, retinitis pigmentosa, and glaucoma. We highlight below the results that demonstrated the possibility that some of these compounds could negatively affect the retina, and therefore, caution should be taken regarding some aspects of its use.

In an evaluation of retinas from the National Toxicology Program bioassay database, Yamashita and co-workers (2016) [59] demonstrated that retinas from a 2-year carcinogenicity study with kava kava extract (KKE) showed a significant increase in degeneration. KKE is derived from the root of the tropical shrub *Piper methysticum*, and it was originally used for ceremonial beverages in the South Pacific. Both males and females exhibited features of retinal degeneration after 0.3 g/kg or 1.0 g/kg KKE dose [59]. In a subsequent study, the group demonstrated that F344N rats dosed with KKE 1.0 g/kg for 90 days did not show signs of retinal degeneration. However, the RPE, only from the superior retina, had a reduced number of phagosomes [60]. The authors speculated that this result could indicate an impairment in photoreceptor outer segment phagocytosis by RPE cells, which could affect the health of the retina after the 2-year exposure observed in the previous study.

Another compound that has been studied due to its potential anticancer properties [61] is hypericin, a polycyclic aromatic naphthodianthrone that occurs naturally. It has been previously shown that hypericin induced cell death of human and bovine isolated RPE cells [62,63]. A more recent study showed that an acute exposure of isolated bovine retinas to hypericin caused a reduction in the amplitude of the b-wave in an electroretinogram recording, indicating an impairment in retinal function.

*Embelia ribes* possess significant potential in the prevention and treatment of several chronic diseases, including arthritis, bacterial infections, cancer, cardiovascular diseases, diabetes, neurological problems, and wound healing [64]. *Hagenia abyssinica* (Rosaceae) is one of the most-used medicinal plants for the treatment of diarrhea and to treat diabetes mellitus in some regions of Africa [65,66]. Post-hatched chicks orally received treatment either with a high dose of 0.25 g (5 g/kg per day) or a low dose of 0.025 g (0.5 g/kg per day) of *Embelia ribes* for 1 or 5 days or *Hagenia abyssinica* for 1 or 9 days. Both compounds impaired visual function (visual discrimination and stimulus detection in the peripheral visual field). High doses of both agents induced degeneration of the ganglion cell layer [67].

Several studies have shown a protective effect of curcumin in different pathological models, such as diabetic retinopathy and ischemia and light-, oxidative stress-, and N-Methyl-D-Aspartate (NMDA)-induced cell death, among others [68,69,70,71,72,73]. However, in one study, curcumin induced apoptosis in mouse-rat hybrid retina ganglion cells, called N18 [74]. Similarly, but with fewer studies, garlic or some of its bioactive compounds show a protective effect in several pathological models. Yet, 24-h exposure to diallyl disulfide (DADS) induced a dose-dependent reduction in N18 cell viability [75]. DADS induced increase in ROS, intracellular calcium and activation of the classic apoptosis mediator, caspase-3 [75].

Concerning natural products, it is important to note the huge number of studies showing protective effects of these substances in animal models of distinct retinal diseases. However, several of these natural herbs/medicines were not tested or fail to show protective effects or, even worse, can be toxic to retinal humans [76,77]. Therefore, clinical studies seem to be extremely important to confirm/reject data from animal models.

## 4. Drugs and Medicine

Despite the BRB, the retina is susceptible to harmful effects of systemic, intravitreal, or topical drugs leading to dysfunction and retinal degeneration. Retinal drug toxicities can be expressed in many ways: disruption of RPE and photoreceptor complex, vascular damage, ganglion cell or optic nerve, cystoid macular edema, crystalline retinopathy, or ganglion cell or optic nerve injury (Figure 2).

### 4.1. RPE and Photoreceptor Complex

The most common presentation of alteration in RPE and photoreceptor complex is pigmentary maculopathy (Figure 3).

#### 4.1.1. Chloroquine Derivatives

Chloroquine (CQ) and its derivative, hydroxychloroquine sulfate (HCQ), are immunomodulatory drugs that are prescribed for malarial prophylaxis and to treat autoimmune conditions such as rheumatoid arthritis or systemic lupus erythematosus. Both medications bind to melanin in the RPE and uveal tissue and can affect metabolic function. Prolonged use of the CQ derivatives typically results in a pigmentary maculopathy (Figure 4). The incidence of retinopathy in patients treated with CQ is approximately 10 to 20% [78].

The classic presentation is a bull’s eye-maculopathy appearance where the fovea is surrounded by a ring of depigmentation followed by a ring of hyperpigmentation (Figure 3). In the advanced stages of toxicity, the pigment abnormalities can involve the peripheral retina with a clinical picture that resembles primary tapetoretinal degeneration with optic disc pallor, retinal vessel attenuation, and bone spicules. Visual acuity is affected when the pigmentary abnormalities involve the center of the fovea. The risk of toxicity is dependent on daily dose and duration of use. At recommended daily doses (maximum of 5.0 mg/kg real body weight for HCQ; 2.3 mg/kg real body weight for HQ), the risk of toxicity up to 5 years is under 1% and up to 10 years is under 2%, but it rises to almost 20% after 20 years. However, even after 20 years, a patient without toxicity has only a 4% risk of converting in the subsequent year. Treatment should be stopped at the first sign of toxicity; otherwise, toxic effects continue to progress despite discontinuation of the drug [78,79].

Interestingly, studies attempting to elucidate the molecular mechanisms responsible for the damage induced by CQ are scarce. The small amount of research focuses on pigmented epithelium cells and leaves the possible actions on neurons aside, especially because CQ-associated retinopathy has as a component of the dysfunction the RPE, such as a breakdown of the permeability barrier. In this sense, Chen et al. demonstrated that CQ treatment reduced proliferation and disorganized cytoskeletal components in RPE1 cells, an immortalized cell lineage of human pigmented epithelium. CQ administered in cultures for 24 h decreased RPE cell number in a dose-dependent manner (10–100 µM) by inhibiting proliferation, but not by cell death. Many cells were in the G0/G1 phase of the cell cycle, while few were present in the S phase, an effect directly related to the reduction in the expression of cyclins (A and E) and phosphorylated-activated cyclin-dependent kinase 2 (CDK2), both proteins related to the progression of the cycle cell to S phase [80]. Simultaneously, CQ treatment increased the number of cells with primary cilium, a structure that is formed during stages of quiescence and stress. Interestingly, CQ induces cell death when the formation of the primary cilium was inhibited with chloral hydrate and ammonium sulfate, indicating that the formation of this structure is favored by the arrest of the cell cycle and works as a protective agent against the toxicity of the CQ [80]. On the other hand, CQ disrupted actin stress fibers and the localization of focal adhesion kinase (FAK) and reduced the nucleation capacity of microtubules by reducing the expression of p150glued protein, a subunit of the dynactin motor complex, but which also acts in the nucleation of the microtubule network [80]. The authors believe that the loss of pigmented epithelium permeability in patients treated with CQ depends on cytoskeletal dysfunction, as found in cultures, which would ultimately contribute to the disorganization of tight junctions. However, evidence to support this hypothesis is lacking.

Unlike RPE-1 cells, which apparently do not suffer cell damage upon exposure to CQ, the ARPE 19 cells die after treatment with this drug [81,82]. Increasing concentrations with CQ (10–100 µM) led to the death of these cells in culture after 24 h of administration, with the same being found in primary cultures of pigmented epithelium of mice. The observed cell death depended on the induction of apoptosis, as seen by the reduction of the anti-apoptotic proteins B-cell lymphoma 2 (Bcl-2) and B-cell lymphoma-extra-large (Bcl-xL) and the increase of the pro-apoptotic proteins BAX and BID [82]. Furthermore, the death of ARPE 19 cells is involved with the inhibition of autophagy. The authors demonstrated a large intracellular vacuolization, which was accompanied by an increase in LC3-A/B, Beclin-1, and p62. The accumulation of autophagy-related proteins was likely due to a weakening of autophagosome–lysosome fusion [82]. The induction of apoptosis and inhibition of autophagy by chloroquine were related to increased phosphorylation of p38 and JNK protein kinases, both responsible for mammalian target of rapamycin (mTOR) phosphorylation. In turn, mTOR induced an increase in the interaction between Beclin-1 and Bcl-2. The interaction between these two proteins causes a mutual inhibition, which explains the effect of chloroquine on apoptosis and autophagy [82]. Interestingly, concomitant exposure with D4476, a casein kinase inhibitor, reversed CQ-evoked effects, demonstrating a key role of casein kinase in controlling apoptosis and autophagy via p38, JNK, and mTOR [82]. Altogether, these works show the neurotoxic mechanisms associated with CQ-induced retinopathy, although little is known about its effects on retinal neurons.

Recently, a pioneering work revealed the cytotoxic action on retinal neurons of rats chronically injected (3 months) with 10 mg/kg of HCQ. After treatment, there was a reduction in the thickness of the GCL and IPL, in addition to a reduction in the number of ganglion cells concomitant with the increase in the death signal in the GCL and in the RPE. These findings correlate with the reduction in rod and cone activity, as measured by electroretinogram, after HCQ treatment [83]. As with chloroquine-treated ARPE 19 cells, HCQ treatment weakened autophagic flux in RGC, but also in cultured Müller cells. Interestingly, HCQ treatment also caused lysosomal dysfunction, which culminated in an imbalance of sphingolipid metabolism homeostasis, leading to a reduction in sphingosine levels and accumulation of C14:0, C18:0, and C22:0 saturated ceramide, toxic bioactive lipids related to cell death [83]. Therefore, the work shows, for the first time, the cell death of retinal neurons by HCQ possibly by weakening of autophagy and accumulation of ceramide.

#### 4.1.2. Phenothiazines

Thioridazine and chlorpromazine are used as antipsychotics to treat schizophrenia and other psychiatric disorders. The exact mechanism of retina lesion is not known, nevertheless it may involve enzyme disruption and abnormal rhodopsin synthesis.

Chlorpromazine toxicity is rare, being more commonly observed with thioridazine due to its piperidyl side chain. Both chlorpromazine and thioridazine accumulate in the melanin of RPE and uveal tissue. In the early stage, there is RPE stippling in the posterior pole; at a later stage, nummular areas of RPE/choriocapillaris loss are seen from the posterior pole to mid-periphery. The late-stage mimics choroideremia or Bietti crystalline dystrophy (BCD), and vascular attenuation and optic atrophy are seen [84,85]. The early fundus abnormalities often advance despite discontinuation of the medication.

The risk of retinopathy is more dependent on total daily dose rather than cumulative amount received. Thioridazine toxicity at dosages less than 800 mg/day is rare, though cases have been reported with lower doses over several years. Regardless of the dose, patients taking thioridazine should be regularly monitored for signs of toxicity [84,85]. A recent study using human retinal organoids, on day 150 of differentiation, showed that thioridazine (135 µM, 24 h) induced a wide range of alterations [86]. Thioridazine induced the expression of genes related to oxidative stress, inflammation, and cell death. Unsurprisingly, an increase in the percentage of cells labeling to chemokine (C-C motif) ligand 2 (CCL2), triggered and released in inflammatory conditions, and to α-crystallin (CRYAB), involved in the protection of cells from stress by binding misfolded proteins, was observed. Classical signals of glia activation were also demonstrated. Accordingly, a widespread reduction of retinal cells (photoreceptors, horizontal, amacrine, Müller glia, and retinal ganglion cells) accompanied by a decrease of approximately 50% in neuroepithelium thickness was shown [86].

#### 4.1.3. Pentosan Polysulfate Sodium

Pentosan polysulfate sodium (PPS) is used for the treatment of interstitial cystitis and is associated with a pigmentary maculopathy. Usually, toxicity is more common in women, after chronic use for over 15 years, and in patients exposed to more than 1500 g [87]. Common symptoms include blurred vision, difficulty reading, metamorphopsia, paracentral scotomas, and prolonged dark adaptation. Signs comprise parafoveal pigmented deposits at the level of the RPE, vitelliform deposits, and patchy paracentral RPE atrophy similar in appearance to pattern dystrophy [87,88,89]. Since PPS is an inhibitor of heparanase, used chronically, a study investigated the consequence of the absence of this enzyme in knockout mice (KO) [90]. Heparanase KO mice at 3-, 9- and 15-weeks-old showed lesions in the retina (central and peripheral), RPE folds, choroidal thickening, cells detached from RPE, increased ONL thickness, and retinal disorganization. The damage in RPE/choroid seemed to progress over time from moderate RPE/choroid changes in 3- and 9-week-old KO mice to severe choroid/RPE protrusions in 58% and 75% for 9- and 15-week-old KO, respectively. There were no signals of proliferation and recruitment of macrophages cells, thus concluding that the RPE protrusions are not related to inflammatory signals from recruited cells.

#### 4.1.4. Clofazimine

Clofazimine is a phenazine dye with anti-mycobacterial and anti-inflammatory action. It binds preferentially to mycobacterial DNA by inhibiting DNA replication and cell growth and is used to treat dapsone-resistant leprosy and autoimmune disorders such as psoriasis and lupus. Ocular side effects include bull’s-eye maculopathy. Drug discontinuation can halt progression but retinopathy does not regress [91].

#### 4.1.5. Deferoxamine

Deferoxamine (DFO) is used as a chelating agent to treat iron toxicity/overload. Signs of toxicity are reticular or vitelliform form abnormalities and/or macular edema due to RPE pump failure. Retinopathy includes several pattern dystrophy-like changes or minimal changes affecting the RPE–Bruch membrane–photoreceptor complex. Multimodal imaging confirms histology findings that photoreceptor outer-derived retinoids, fluorophores, and RPE displacement or clumping are entangled in DFO retinopathy, leading to unequivocal RPE atrophy in many cases of pattern dystrophy–like changes. Drug cessation can reverse established mild retinopathy. Nevertheless, when exposure is prolonged, RPE and outer retina damage may persist [92]. Nonetheless, iron chelant, including deferoxamine, has been shown to protect retinal cells in different degenerative models, such as NMDA-induced excitotoxicity [93], cell death promoted by oxidative stress in ARPE 19 cells [94,95], or 611 photoreceptor lineage cells [96] by blocking ferroptosis.

#### 4.1.6. Antiretroviral Therapies

Even though antiretroviral drugs can help arrest human immunodeficiency virus (HIV) progress or other infection-associated retinal disease, an undesirable rare retinal toxicity could occur [97]. Didanosine (DDI) is a nucleotide reverse transcriptase inhibitor used to treat individuals with acquired immunodeficiency syndrome (AIDS), probably because it inhibits polymerase (pol-γ), the enzyme responsible for replication and repair of mitochondrial DNA. DDI can cause mitochondrial dysfunction and toxicity resulting in damage to the optic nerve and RPE; peripheral field loss occurs with concentric loss/mottling of RPE (areas of chorioretinal atrophy), beyond arcades to mid-periphery, bilaterally symmetrical [98] (Figure 5).

#### 4.1.7. Mitogen-Activated Protein Kinase Inhibitors

Mitogen-activated protein kinase (MEK) inhibitors (trametinib (Mekinist), cobimetinib (Cotellic), binimetinib (Mektovi), and selumetinib (Koselugo)) are used to treat metastatic melanoma. The toxicity mechanism is thought to be due to RPE-induced dysfunction with subsequent accumulation of subretinal fluid. The most common structural abnormality found is bilateral multifocal serous retinal detachment with at least one focus involving the fovea. Onset can occur shortly after initiation of therapy. Visual symptoms are typically minimal with the fluid often spontaneously resolving. For persistent cases, discontinuation of the drug usually leads to complete resolution [99].

Human neuroretina shows phosphorylated ERK, which is inhibited by binimetinib treatment [100]. Phosphorylated ERK status is recovered after ceasing MEK inhibitor treatment both in ARPE 19 cells and primary neuroretina cells from human tumor eyes [100]. These results indicate that the disappearance of retinopathy with the discontinuation of binimetinib treatment is associated with the reactivation of ERK.

A study with retinal detachment (RD) mouse model showed that selumetinib prevented the increase in phosphorylated ERK in Müller glial cells, whereas it did not block the Müller reactive signals and photoreceptor death induced by RD [101]. The treatment also blocked the increase in the number of Iba1-positive cells promoted by RD.

#### 4.1.8. Fibroblast Growth Factor Receptor Inhibitors

Fibroblast growth factor receptor (FGFR) inhibitors are chemotherapeutic agents used to treat cholangiocarcinoma and urothelial carcinoma [102]. Currently, FDA-approved drugs include erdafitinib, pemigatinib, and infigratinib [103]. FGFR inhibitors appear to cause serous retinal detachments, similar to MEK retinopathy. Few reports are available in the literature of FGFR retinopathy [104].

#### 4.1.9. Sildenafil

Sildenafil is used to treat erectile dysfunction and pulmonary artery hypertension. It blocks phosphodiesterase 5, an enzyme that promotes breakdown of cyclic guanosine monophosphate (cGMP). A possible side-effect is dilatation of choroidal vasculature and secondary serous macular detachment, and retinal vascular occlusion can occur [105].

The effect of sildenafil (225 μM treated for 7 days) was evaluated in retinal organoids, derived from human pluripotent stem cells, after 150 days of differentiation [86]. Sildenafil significantly reduced the organoid neuroepithelium thickness, affecting primarily photoreceptors. Although blue cones were found only rarely in organoids, sildenafil significantly decreased the number of this cell type. The phototransduction pathway was also affected after sildenafil treatment, resulting in gene expression changes.

Mice treated with sildenafil showed a reversible increase in maximal retina vessel dilatation and choroid effusion promptly after intravitreal injection and 30 min after intraperitoneal injection [106]. In 5% of mice, sildenafil provoked RGC loss and damage of optic nerve after 21 days of the treatment. In an in vivo approach to evaluate the effect of sildenafil in the mouse retina, QUEnch-assiSTed (QUEST) magnetic resonance imaging (MRI) was used in subretinal space using QUEST optical coherence tomography (OCT), while QUEST optokinetic tracking (OKT) was used for cone-based vision [107]. QuestMRI showed an increase in oxidative stress in a group treated with sildenafil when compared to a group exposed to sildenafil plus antioxidants. This effect was only detected in the peripheral superior retina [107]. Levels of ROS evaluated by DCF staining in freshly isolated retinal sections were higher all over the retina treated with sildenafil compared to saline, with more prominent labeling in the superior retina. ONL thickness was constant regardless of treatment with sildenafil or antioxidants. At 5 h post-sildenafil treatment, contrast sensitivity was significantly lower-than-normal and similar even in the presence of antioxidants.

Finally, by using a multistep light stimulus electroretinography, it has been reported that sildenafil affects retinal rod function in female rats [108]. The records were performed 1.5 h after treatment at doses of 15, 50, and 150 mg/kg. A decrease as well as an extension in the amplitude and implicit time of the a-wave (≥50 mg/kg) and of the b-wave (all doses) were observed.

#### 4.1.10. Cisplatin

Cisplatin (cisdiamminedichloroplatinum—CIS) has been used effectively for years as a chemotherapy drug in the treatment of solid tumors, metastases, and small cell cancers with unknown primary tumors [109]. Despite its success in inducing tumor death and remission of associated symptoms, several studies report side effects related to the visual system in patients treated with CIS. In these cases, partial loss of vision, retinal detachment, thinning of the optic nerve fibers, changes in the electroretinogram, occlusion of the middle retinal, and cilioretinal arteries have already been demonstrated [110,111,112,113]. At the cellular level, the cytotoxicity of CIS was confirmed in ocular tissue cells of different species. In rat retinas, a single intraperitoneal injection with CIS (16 mg/kg) increased the levels of MDA, as well as decreased the levels of reduced glutathione (GSH). Immunohistochemical assays revealed an increase in labeling for 8-hydroxy-2p-deoxyguanosine (marker of DNA damage) in horizontal cells and for endothelial nitric oxide synthase (eNO) in retinal blood vessel cells [114]. Blood samples from rats administered for 14 days with CIS (2.5 mg/kg) also revealed an increase in MDA, myeloperoxidase (MPO), and the levels of the total oxidant system, but also in the pro-inflammatory cytokines, including the tumor necrosis factor alpha (TNF-α) and interleukin-1β (IL-1β). At the same time, there was a reduction in GSH, SOD activity, and levels of the total antioxidant system [115,116,117]. After treatment with CIS, the retinal tissue showed degeneration, edema, and vascular congestion with disorganization of the retinal layers. Optic nerve tissue anomalies have also been observed, such as destruction, hemorrhage, edema, and an increase in the number of astrocytes and polymorphonuclear leukocytes [114,115,116,117]. These data indicate that the ocular disorders found in patients treated with CIS may be due to an imbalance of antioxidant defenses and the production of ROS and nitrogen species, which induces cell damage and an inflammatory process. This premise can be supported by considering that antioxidant agents such as astaxanthin, rutin, coenzyme Q10, and lutein are able to reverse retinal dysfunctions related to CIS toxicity [114,115,116,117,118]. Interestingly, CIS treatment may be activating parallel signaling pathways that favor DNA damage response and cell survival [119]. The genotoxic stress triggered by CIS treatment induces the formation of the primary cilium, a structure related to DNA damage, from the centrosome in a p53 protein-dependent manner in RPE1 cell cultures [119]. Whether CIS-dependent ciliogenesis occurs through these two mechanisms is a question that still remains unanswered but opens up more possibilities for the search for its molecular actions.

### 4.2. Retinal Vascular Damage

Many drugs can damage the retinal vasculature, by inducing a hypercoagulable state, or by particle clogging of blood vessels. Some might be used as intraoperative ocular medications as aminoglycosides, moxifloxacin, or vancomycin.

#### 4.2.1. Talc

Talc retinopathy is characterized by the presence of small, yellow crystals located in small retinal vessels and within different retinal layers. Ocular damage usually develops after chronic intravenous drug abuse and manifestations range from asymptomatic crystalline retinopathy to severe ischemic manifestations of capillary non perfusion. The presence of crystals is thought to be secondary to emboli derived from talc, which is an insoluble inert particulate filler material used in some oral (methylphenidate hydrochloride, methadone, pentazocine, and amphetamine), inhaled (crack cocaine), and intravenous (cocaine and heroin) preparations.

#### 4.2.2. Interferon

Interferons are classified into three major types—INF-α, INF-β, and IFN-γ—and have been used for treatment of different pathologies, including Kaposi sarcoma, hepatitis B and C, multiple sclerosis (MS), and malignant osteopetrosis. Systemic therapy has been associated with retinal vasculopathy characterized by cotton wool spots, intraretinal hemorrhages, microvascular changes including capillary drop-out, CME (typically in the posterior pole and peripapillary region), venous occlusion, or arterial occlusion, consistent with ischemic retinopathy [120]. The exact mechanism of toxicity is not known but may involve impairment of retinal microcirculation. Changes typically present 4–8 weeks after initiation of therapy and usually regress after treatment cessation [120].

Here we focus on studies that shed light on these molecules as central regulators and not just inflammatory markers. Among these three types of interferon, only IFN-γ was related to possible retinal injurious effects. Roche and collaborators (2018) reported that IFN-γ released by microglia induces pSTAT3 signaling in Müller cells and increases glial fibrillary acidic protein (GFAP) expression [121]. These results were explored in the context of the retinitis pigmentosa (rd10) mouse model, which increased GFAP staining throughout time. IFN- γ was also linked to the increase of BRAF-activated non-coding RNA (BANCR), a long non-coding RNA involved in the inflammatory context of RPE dysfunction associate to diseases like AMD, acting through signal transducer and activator of transcription 1 (STAT1) phosphorylation in ARPE-19 cells [122]. This increase was progressive, reaching up to 30 times higher concentration of the RNA when treated with 100 units/mL of IFN-γ. In addition, in ARPE-19 cells and in this context of inflammation and AMD progression, IFN-γ (50 ng/mL–48 h) induced cell death [123]. The authors suggested that ferroptosis plays a role in this cell death since Fe^2+^ is increased by five times after a 48h treatment, alongside oxidative stress and GSH depletion. Increased Fe^2+^ levels were observed as a consequence of the downregulation of the Fe^2+^ efflux protein SLC7A11 close to 50%, and oxidative stress was associated with the aforementioned Fe^2+^ changes alongside downregulation of system xc^-^ through the activation of the signaling pathway involving janus kinase enzyme 1/2 (JAK1/2) and STAT1. Similar results were obtained in the retina of mice treated intravitreally with IFN-γ (1 ng/μL for 7 days), and ferroptosis inhibitors can rescue retinal degeneration after NaIO_3_ treatment as a model of AMD, strengthening the hypothesis that this pathway could be important in AMD and other eye conditions that have an inflammation component. Additionally, IFN-γ may contribute to hyperglycemia-induced reduction of platelet endothelial cell adhesion molecule-1 (PECAM-1), as a treatment of rat retinal microvascular endothelial cells (RRMECs) with IFN-γ (30 ng/mL) reduces PECAM-1 levels by 57%, possibly through increased ubiquitination and degradation. PECAM-1 is a surface immunoglobulin involved in endothelial cell-to-cell adhesion, maintenance of blood barrier, and leukocyte transmigration, among others, in RRMECs [124].

Controversially, in a context of diabetic retinopathy, diabetic mice lacking IFN-γ showed a more than two-time increase in the mRNA expression of vascular endothelial growth factor (VEGF), intercellular adhesion molecule 1 (ICAM-1), retinoic-acid-receptor-related orphan nuclear receptor gamma (ROR-γt), a transcription factor of immune cells T helper (Th) 17 cells, and a more than ten-times increase in splenic IL-17-producing CD4^+^ cells in comparison to diabetic mice [125]. Meanwhile, IFN-γ shows an anti-angiogenic effect in a mouse model of oxygen induced retinopathy (OIR) [126]. These data show a role for IFN-γ as a critical regulator of inflammation and a possibility to explore it as a tool given its role as an inflammatory cytokine regulated in eye disease.

#### 4.2.3. Ergot Alkaloids

Ergot alkaloids are mycotoxins produced by many fungal species of the Claviceps genus. There are four main types of ergot alkaloids: clavines, lysergic acids, lysergic acid amides, and ergopeptides. One of these ergopeptides is dihydroergotamine, which has been extensively used in the treatment of migraine. The antimigraine effect is mainly related to its agonist activity at 5-hydroxytryptamine receptor 1B (5-HT1B), 5-hydroxytryptamine receptor 1D (5-HT1D), and 5-hydroxytryptamine receptor 1F (5-HT1F) receptors [127]. One case report described that one patient that received for the first time the oral medication Cefalium, a medication used to treat migraine, which contains dihydroergotamine in the formula, presented some ocular anomalies such as acute bilateral transient myopia, retinal folds, and island of choroidal delay after one day of treatment. The interruption of the treatment was able to solve all clinical symptoms [128].

Dopaminergic agonists derived from ergot are a group of drugs consisting of bromocriptine, cabergoline, dihydroergocryptine, lisuride, and pergolide. They have been available on the market for many years and are mainly used to treat Parkinson’s disease, either alone or in combination with other medicines. A recent report described that knockout mice of rod transducin G protein subunit alpha transducin 1 (Gnat1), visual arrestin 1 (ARR1), or rhodopsin kinase 1 (GRK1) showed light damage and robust retinal inflammation after bright light exposure [129]. The pretreatment with metoprolol plus tamsulosin and bromocriptine protected the retina in all genetic knockout mice [129]. Abnormalities of angiogenesis are very common in age-related macular degeneration and proliferative diabetic retinopathy. In a study with a zebrafish animal model, retinal neovascularization induced by cobalt chloride promoted hypoxia. Pre-incubation with bromocriptine, cabergoline, pergolide, and all ergot-derived D2 dopamine receptor agonists significantly inhibited abnormalities of angiogenesis, decreasing mRNA expression levels of vascular endothelial growth factor Aa (VEGFAA) [130].

Lysergic acid diethylamide (LSD) is a potent synthetic psychedelic drug that can be derived from the ergot alkaloids. Visual changes are some of the effects after LSD use. One study investigated the effect of LSD on macrophage activation state and its toxicity to photoreceptor cells in vitro. They showed that the treatment of macrophage cultures with LSD induced a change to a pro-inflammatory profile [131]. LSD treatment of co-cultured macrophages with photoreceptors induced an increase in the oxidative stress markers and toxicity on photoreceptor cells [131]. Another study demonstrated that C57BL/6 mice treated with LSD had a decrease in electroretinography response and the loss of photoreceptor cells. This cell death of photoreceptors was mediated by upregulation of p-JAK1/p-STAT1 pathway [132].

#### 4.2.4. Gemcitabine

Gemcitabine, a pyrimidine nucleoside analog, is a chemotherapy drug used as a treatment for different types of cancer, including bladder and breast cancer. Some studies have reported retinopathy associated with the use of this chemotherapeutic agent. One study demonstrated that one patient had several issues related to use of gemcitabine, such as a decrease in vision with appearance of cotton wool spots and intraretinal hemorrhages [133]. Another case report described that one patient had macular infarction after chemotherapy with gemcitabine and carboplatin [134]. More recently gemcitabine-associated retinal pathologies, such as presence of bilateral peripheral exudative retinal detachment, retinal edema, and Elschnig’s spots, were also described [135].

### 4.3. Cystoid Macular Edema

Cystoid macular edema (CME) may occur after the treatment with fingolimod Gilenya for MS, topical prostaglandin analogs (e.g., latanoprost) for ocular hypertension or glaucoma, nicotinic acid (niacin) for lipid disorders, and/or paclitaxel treatment or DFO for iron toxicity/overload. Drug cessation results in resolution of the CME, although topical and local steroids or topical non-steroidal anti-inflammatory drugs have been used to facilitate resolution.

#### 4.3.1. Epinephrine

Epinephrine, an endogenous molecule that can be used to treat cardiac arrest and anaphylaxis in a hospital environment, may also influence retinal physiology. Systemic adrenergic stimulation with isoproterenol, a β1-and β2-adrenergic receptors agonist, impact RPE renin expression, with implications on retinal pathophysiology [136]. The influence of stimulating the adrenergic system in the retina is not only related to changes in the renin–angiotensin system, but the stimulation can also affect RPE ion transport. Treatment with epinephrine induced a small, but rapid increase in what the authors called “short-circuit current” (current required to reduce the potential across the epithelial membrane to zero) [137]. Angiographic changes such as choroidal vessel dilation, increase on choroidal thickness, disruption and effacement of the ellipsoid zone, and elongation and protrusion of photoreceptor outer segments have been reported after epinephrine treatment for 8 weeks in cynomolgus monkey [138].

#### 4.3.2. Nicotinic Acid/Niacin

Nicotinic acid, also known as vitamin B, and its derivatives such as nicotinamide, can influence retinal and RPE metabolism. For instance, there is an established method for culturing ARPE-19 cells that uses nicotinamide to stimulate cell growth and differentiation [139].

#### 4.3.3. Paclitaxel and Docetaxel

Paclitaxel and docetaxel are antineoplastic agents of the taxane class of drugs used in the therapy of many solid tumors, including breast and lung cancer. They act by promoting and stabilizing microtubule assembly, while preventing physiological microtubule depolymerization/disassembly in the absence of GTP. This leads to a significant inhibition of cellular mitosis and cell death [140]. There are several case studies reported in the literature linking retinopathies, such as phototoxic maculopathy and cystoid macular edema, induced by paclitaxel and docetaxel [141,142,143,144], but there are few reports about studies with these compounds in animal models or in vitro models. In C57BL/6J littermate pups, it was seen that paclitaxel treatment was able to reduce the number of retinal vascular branches in a dose-dependent manner during mouse retinal development in vivo [145]. Another study in a rat model showed that a single intraperitoneal injection of paclitaxel led to an increased retinal vascularity and rosette-like structures in the outer nuclear layer, a lesser number of astrocytes and oligodendrocytes, and some signs of cellular necrosis [146].

### 4.4. Crystalline Retinopathy

#### Tamoxifen

Tamoxifen is a selective estrogen receptor modulator and has been used to treat breast cancer. Retinopathy induced by tamoxifen is characterized by crystalline deposits and pseudocystic foveal cavitations. These findings are like macular telangiectasia type 2, suggesting a similar pathogenesis involving Müller cell dysfunction. Tamoxifen is associated with thinner choroid and total retinal thickness, suggesting that there were structural changes in patients without symptoms that could be early signs of RPE and photoreceptor damage. Toxicity is dependent on dose and length of use and typically manifests after 2-3 years. Visual function and macular edema typically improve after drug cessation, though the crystalline deposits remain [147,148] (Figure 6).

Recent in vitro data confirms RPE susceptibility to tamoxifen treatment by decreasing primary human fetal RPE cells (H-RPE) and ARPE-19 human RPE cells viability to less than 10% in a concentration-dependent manner [149]. The authors also investigated possible mechanisms triggering cell death. Tamoxifen increased total ROS and superoxide-positive cells as well as the autophagy marker LC3B II/I ratio, which could be reversed, respectively, by reducing oxidative stress with NAC, or autophagy with Baf-1, maintaining cell viability at control levels. Furthermore, NAC prevented LC3B II/I ratio increase, suggesting that tamoxifen promotes ROS-induced autophagy cell death. Additionally, tamoxifen increased inflammatory markers IL-1β, NOD-, LRR- and pyrin domain-containing protein 3 (NRLP3), and apoptosis-associated speck-like protein containing a CARD (ASC). Anti-inflammatory and immunomodulatory molecules reverse the effects of tamoxifen, indicating a possible therapeutic tool [149].

Despite the recent in vitro data on RPE cells, in which tamoxifen is harmful, previous results have shown a protective role for the estrogen receptor modulator. Wang and collaborators (2017) observed that chronic tamoxifen (500 mg/kg—on diet) treatment prevented the effects of a mouse model of retinal light injury (LI) after 7 days, restraining the LI-induced retinal thickness decrease to almost zero, diminishing retinal detachment to less than 10% chance when compared to control, and normalizing ERG parameters. It also exerted a specific effect on rescuing the thickness of the outer retina, where the photoreceptors are located, correlating the results with diseases that promote photoreceptor degeneration, such as AMD, diabetic maculopathy, and retinitis pigmentosa. The protective effect was associated with more than 50% reduction in microglial activation and cytokine release promoted by the tamoxifen diet after 7 or 14 days of LI [150].

### 4.5. Damage to Ganglion Cell Layer or Optic Nerve

#### Methanol

Methanol intoxication is a very debilitating condition, most commonly found in developing countries where ingestion of contaminated alcohol in beverages could lead to a variety of systemic symptoms that range from mild intoxication, such as abdominal pain, nausea, vomiting, headache, general weakness, dyspnea and nervous system disturbances, including the retina, to more severe intoxication leading to renal failure, cardiovascular alterations, rhabdomyolysis, convulsions, coma and eventually death [151]. Visual symptoms might appear hours after ingestion, and they vary widely from progressive decrease in vision to dyschromatopsia, scotoma, and photophobia, accompanied by hyperemic and edematous optic disk acutely, while atrophy and pallor can be observed chronically [152]. The molecule itself is not the most alarming toxic agent, but rather, its metabolite, formic acid, represents a serious threat, being able to bind and inhibit cytochrome c oxidase, an enzyme of the mitochondrial respiratory chain, and therefore, inhibiting oxidative phosphorylation, causing ATP deficiency, and increasing oxidative stress, damaging a series of cellular components [151].

In animals models it has been shown that moderate (3%) to high (4%) methanol levels in the water can disrupt the retina in zebrafish ex utero model of developing retina [153]. The methanol exposure led to small eye phenotype, morphological abnormalities in retinal pigmented epithelium and photoreceptors, and inhibition of differentiation and proliferation. In adult rats, daily i.p injections of methanol (50% wt/vol) for 7 days induced a 10% reduction of RGCs, and in the occipital cortex, a 10-times increase in caspase-3+ cells, 20% increase of glial cells (GFAP+), 46% decrease of brain-derived neurotrophic factor (BDNF), and a 54% increase in serum levels of NO, which further increases methanol-induced damage [154]. Interestingly, LED photoirradiation (670 nm) could reverse these effects, suggesting an elegant non-invasive possibility of therapy that affects not only the eyes. A decrease in RGC density and thickness alongside increased caspase-3 was also observed in a rat orally administrated methanol-induced toxicity model, an effect reversed by citicoline (1 g/kg/day) [155]. Very recently, Dorgau and colleagues (2022) aimed to establish a retinal organoid model to study drug toxicity and contribute to the development of treatment strategies [86]. The retinal organoids assemble from human pluripotent stem cells (hPSCs) and are arranged in a layered structure, expressing biomarkers for key cell types. Methanol treatment (32 mM–24 h) reduced the staining for middle/long wavelength cone opsin to almost zero, and although it did not change expressing markers for ganglion cells, it did reduce the number of active RGCs by half in response to white light pulses.

Moreover, in a rat methanol-induced toxicity study (3 g/kg—oral, for 7 days), Rutin (3,3′,4′,5,7-pentahydroxyflavone-3-rhamnoglucoside) or taxifolin (3,3′,4′,5,7-pentahydroxiflavanone), flavonoids found in vegetables and fruits, given after methanol ingestion, could prevent methanol induced increase in inflammatory markers (IL-1β, NF-κB and TNF-α) and oxidative stress markers (8-OHdG, MDA, MPO) in the optic nerve, alongside restoration of histologic pattern and avoidance of edema, hemorrhage, and congestion, suggesting an attenuation of optic neuropathy [156,157].

## 5. Conclusions

The retina is a quite complex tissue with a critical role for vision. Thus, it is important to be aware of possible side effects of substances that can reach the retina. This review gives information on several chemical agents that can lead to retinal malfunction or damage.

The data in the literature is clear about the deleterious effect of pesticides and calls for maximal restriction of the use of these substances. Glyphosate, the most-used herbicide in the United States and Brazil, downregulates genes important to eye development and causes several alterations in the eye. Chlorpyrifos, an insecticide banned from Europe but still used in the USA and Brazil, provokes an array of cellular alterations in the retina, such as oxidative stress and apoptosis, and induces morphological changes. Thiamethoxam and lefenuron are also insecticides that affect eye and retinal function. The insecticide cypermethrin also induces apoptosis of retinal cells. Finally, the fungicides triphenyltin and thiram impair eye and retinal development as well as provoke retinal detachment, hemorrhage in the fundus of the eye, and miosis.

It is also unambiguous that treatments for a wide range of diseases, such as malaria, autoimmune conditions, psychiatric disorders, different types of cancer, and migraine, among many others, can cause alterations in retinal function, which could lead to visual impairment or more drastically, retinal cell death. The side effects vary from injuries mainly in RPE and photoreceptor complex, retinal vascularization, and ganglion cell layer or optic nerve to cystoid macular edema as well as crystalline retinopathy.

Moreover, although animal models are an important tool to understand the molecular and cellular mechanisms that underlie these toxic effects, for many medicines/drugs, no recent studies were found. For medicines/drugs that mainly affect RPE and photoreceptor complex only a few studies were performed, particularly more abundant for chloroquine derivatives. Nevertheless, the available data focus on the ability of these agents to induce cell death. Similarly, except for IFN-γ, there are no molecular and cellular studies for medicines/drugs (Talc, Ergot alkaloids and gemcitabine) that mostly affect retinal vascular components. Likewise, there are few studies investigating the molecular mechanisms of medicines/drugs that are predominantly related to cystoid macular edema or crystalline retinopathy. Most of the data show involvement of oxidative stress, increase in the release of inflammatory agents, and gliosis. These data highlight the importance and the challenges of continuous research to better understand the mechanisms involved in the toxicity of these medicines/drugs. Furthermore, the present review highlights the importance of research that aims to discover new drugs, with fewer or no harmful effects to the retina.

Concerning natural products, there are many studies showing a protective effect in different animal models of retinopathies. However, fewer studies evaluate the effect of these substances in the retinal physiology. Kava kava extract, *Embelia ribes*, and *Hagenia abyssinica* can impair visual function and induce retinal degeneration. On the other hand, the number of studies investigating the outcome of many other natural agents, especially the combination of herbs (mix) as used by people, in the retina are still low. As mentioned before, even though there is plenty of evidence for neuroprotective effects of several natural herbs in animal models, there are few studies investigating the outcome in the human retina. Besides that, among the published studies with humans in the field, some agents fail to show protective effects or, even worse, are deleterious to the human retina. Therefore, clinical studies seem to be extremely important to confirm/reject present data from animal models.

Finally, but not less important, it is crucial to call attention to the extremely low number of animal model studies using females. One study showed that 80% of drugs withdrawn from the USA market (from 1997 to 2000) occurred due to a greater risk for women than men [158]. Therefore, it is very dangerous to try to translate these studies to the human population without preclinical studies in female animals.

## Figures and Tables

**Figure 1 ijms-23-08182-f001:**
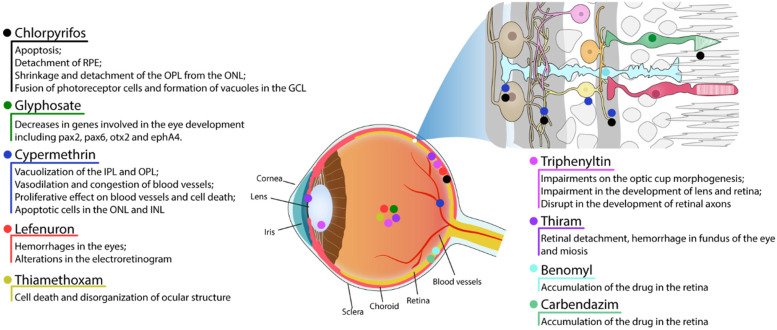
Schematic figure showing the areas affected by the pesticides. Effects in retinal vascularization and RPE are showed in the eye draw whereas in neural retina are represented in retinal scheme.

**Figure 2 ijms-23-08182-f002:**
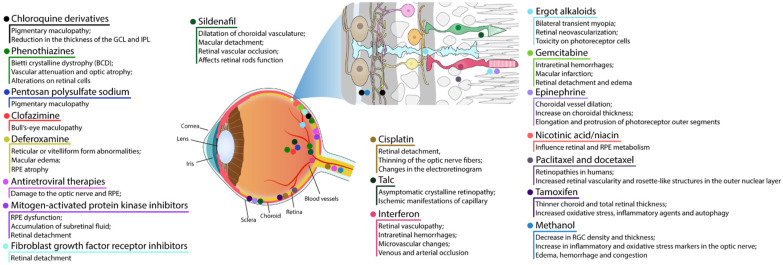
Schematic figure showing the areas affected by some drugs. Alterations in retinal vessels and RPE are represented in the eye whereas impact in retinal cell are showed in the retinal scheme.

**Figure 3 ijms-23-08182-f003:**
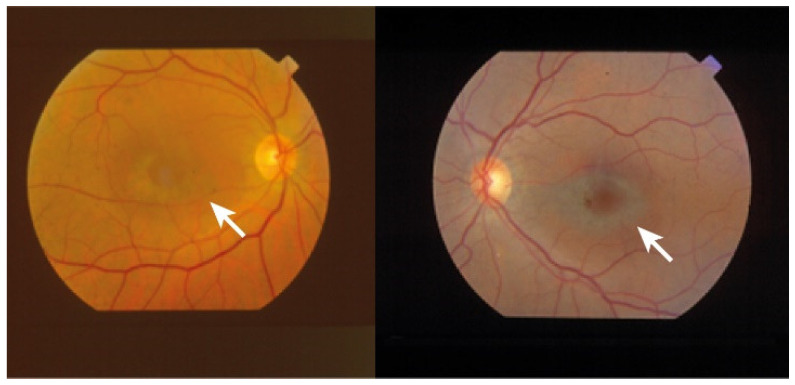
Fundus image: white arrow shows a concentric epithelial degeneration around the fovea, called bull’s eye maculopathy.

**Figure 4 ijms-23-08182-f004:**
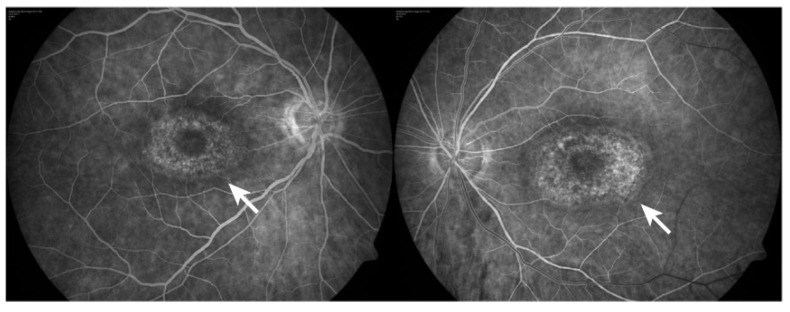
Hydroxychloroquine toxicity: white arrow shows presence of parafoveal zone of hyperfluorescence on fluorescein angiography.

**Figure 5 ijms-23-08182-f005:**
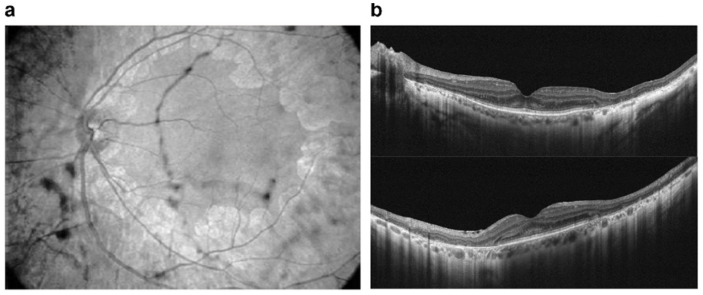
Multimodal imaging: (**a**) infrared light retinography shows confluent plaques of chorioretinal atrophy in posterior pole, with the macular area more preserved; (**b**) OCT shows marked impairment of the outer neurosensory retinal layer, retinal pigmented epithelium, and perifoveal choroid. Diffuse thinning is seen with marked atrophy of the outer nuclear layer, external limiting membrane, and ellipsoid zone. Diffusely increased posterior light reflectance demonstrates associated retinal pigmented epithelium atrophy. The foveal area maintains normal architecture but shows slightly increased internal reflectivity of the outer nuclear layer.

**Figure 6 ijms-23-08182-f006:**
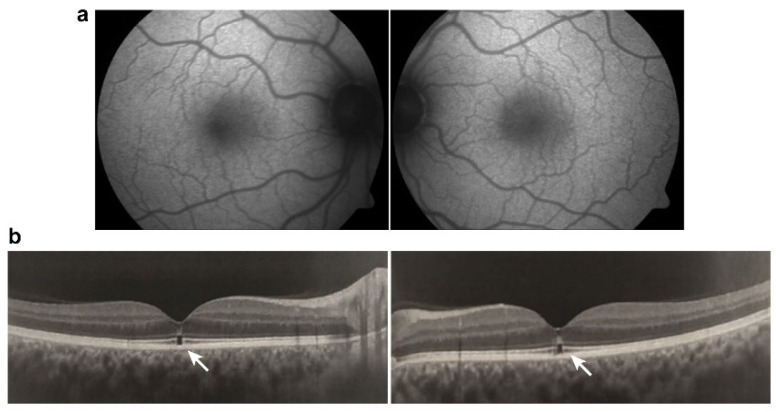
Tamoxifen retinopathy: (**a**) fundus autofluorescence image showing no abnormality; (**b**) optical coherence tomography: white arrow demonstrates ellipsoid foveal zone disruption and external limiting membrane hyperreflectivity.

**Table 1 ijms-23-08182-t001:** Pesticides and their actions.

Chemical Category	Pesticide	Action
Organophosphates	Chlorpyrifos	Inhibit AChE
	Glyphosate	Inhibition of the enzyme 5-enolpyruvylshikimate-3-phosphate synthase
Pyrethroid	Cypermethrin	Data not available
Benzoylurea	Lefenuron	Data not available
Neonicotinoid	Thiamethoxam	Data not available
Organotin	Triphenyltin	Agonist of retinoid X receptors
Organosulfur	Thiram	Data not available
Benzimidazoles	Benomyl	Data not available
	Carbendazim	Data not available ^1^

## Data Availability

Not applicable.

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
