# Peer review of "Retinal Toxicity Induced by Chemical Agents"

_ijms, 2022, doi:10.3390/ijms23158182_

Round 1
Reviewer 1 Report
This manuscript entitled “retinal toxicity induced by chemical agents” reviews the impact of the exposure to several chemical agents found in pesticides, natural products such as medicinal herbs on the human eye, characterized by retina neurotoxicity. This review is original, gathers a corpus of 161 references and of sufficient quality for submission to International Journal of Molecular Science. The following minor revisions should be considered before publication:
1. The introduction could be shortened, and written in a less lyrical way, e.g. the first paragraph. Line 62 : “The axons …nuclei” : sentence unclear, please rewrite. Line 114 : “At least…isolation” : references are needed to back these claims.
2. The rationale of this review could be better defined. The substances considered in this study could be clearly mentioned in the introduction and their choice justified.
3. A schematic showing more clearly the relationship between exposure to a substance (long-term/acute, substance concentration) and the possible damage to the eye (alteration in photoreceptor complex, retinal vascular damage, damage to GCL) could be added past the introduction section.
4. A table of contents could be added to the manuscript.
5. While the evidence of the relationship between retina damage and exposure to a given substance is of great interest, a further insight could be given with regards to a quantitative measure of the exposure considered, if possible.
6. The conclusion section needs rewriting, in a more factual way, without references as those claims should be moved to an introduction or discussion section. The conclusion should be written as a summary of the findings in this review, followed by an opening towards future outcomes.
Author Response
Dear reviewer,
Please see the attached file with the responses for your comments.

Reviewer 2 Report
Dear authors,
your review is comprehensive and includes various groups of artificial and natural substances affecting the retina. I have no doubt that you have carefully looked through the databases and collected data.
What I miss in the first part about pesticides is some brief outline. For example, the one you have in part 4. Drugs and medicine. In that part you focused at type of damage and then at the chemical/drug group.
In part 2. Pesticides, you only listed the names of pesticides/active substances, or placed them in the main group (insecticide/fungicide). These chemicals should be categorized because substances of a certain group usually have the same mechanism of action. And it is assumed that other substances of this group will also have a negative effect on the retina.
My proposal is to reorganize the part 4. Pesicides according to the category of chemical substances, not only insecticide, but e.g. organophosphates, neonicotinoids, pyrethroids.
Secondly, rewrite the part about chlorpyrifos to make it shorter. It is too lengthy and does not focus on the most essential findings, which are hidden under a tangle of other information.
Author Response

(The authors gave the same response as above.)
